# Synapsins and the Synaptic Vesicle Reserve Pool: Floats or Anchors?

**DOI:** 10.3390/cells10030658

**Published:** 2021-03-16

**Authors:** Minchuan Zhang, George J. Augustine

**Affiliations:** Lee Kong Chian School of Medicine, Nanyang Technological University, Singapore 308232, Singapore; minchuan001@e.ntu.edu.sg

**Keywords:** synapsins, synaptic vesicle trafficking, neurotransmitter release, presynaptic terminals

## Abstract

In presynaptic terminals, synaptic vesicles (SVs) are found in a discrete cluster that includes a reserve pool that is mobilized during synaptic activity. Synapsins serve as a key protein for maintaining SVs within this reserve pool, but the mechanism that allows synapsins to do this is unclear. This mechanism is likely to involve synapsins either cross-linking SVs, thereby anchoring SVs to each other, or creating a liquid phase that allows SVs to float within a synapsin droplet. Here, we summarize what is known about the role of synapsins in clustering of SVs and evaluate experimental evidence supporting these two models.

## Introduction

Exocytosis of neurotransmitters stored in synaptic vesicles (SVs) mediates synaptic transmission between neurons. In the central nervous system, typically tens to hundreds of SVs are clustered within a presynaptic bouton [1,2] (Figure 1A). These SVs can be classified into different pools, based on their availability to participate in exocytosis and, secondarily, according to their spatial distribution [2,3] (Figure 1B). SVs that are docked at the active zone (AZ)—the site of exocytotic discharge of SV contents into the synaptic cleft—are thought to rapidly fuse with the plasma membrane in response to a presynaptic action potential and thus form a readily-releasable pool (RRP) [3,4,5]. A large SV cluster distal to the AZ is believed to serve as a store that replenishes SVs into the RRP following exocytosis, thus called the reserve pool (RP) [6,7]. The remaining SVs that are neither docked in the RRP nor trapped in the RP are assigned as a recycling pool, because they involve SVs that have recently been recycled from the plasma membrane following exocytosis. Recycling SVs can intermix with RP vesicles and can be re-released by becoming part of the RRP or RP [3]. The relative fraction of SVs in each pool differs between synapses and is dynamically regulated, varying in response to many conditions such as during neural development [8] and in response to synaptic plasticity [9,10]. In general, most SVs (80–90%) are found within the RP, with only a small fraction of SVs in the RRP (1%) or the recycling pool (10–15%) [3,4,11]. Because the relatively small number of SVs in the RRP are rapidly depleted during repetitive synaptic activity, movement of SVs from the RP to the RRP broadens the bandwidth of synaptic transmission by enabling transmission to be sustained during high-frequency activity [12,13]. A recent study suggests that such movement also is an important mechanism for forming short-term memories [10]. Impairing the RP is associated with maladaptive changes in circuit activity, such as tremor and epilepsy [14,15,16], perhaps caused by disturbing the balance between synaptic excitation and inhibition [14,15,17,18].

Although the RP concept has been around for many years, we still do not understand how the RP is made and how SVs are translocated from it to the RRP. A fundamental clue comes from studies of *synapsins*, which are SV proteins found in all presynaptic terminals. Perturbation of synapsins causes a dramatic reduction in the number of SVs within the RP, particularly at excitatory glutamatergic synapses [6,19,20]. Here we will briefly summarize experimental evidence that links synapsins to maintenance of the RP and provide a critique of current hypotheses to explain how synapsins keep SVs within the RP.

## Synapsins Bind to SVs

Synapsins are a family of peripheral proteins that bind to the SV membrane. They are encoded by three genes: *SYN 1, 2, 3*. Alternative splicing results in a/b protein variants for each synapsin gene (and even more for *SYN 3*), yielding five major members in the family (Figure 2A). These isoforms share three conserved domains: domain A (residues 1 to 28 for synapsin 1, or 29 for synapsin 2 in rat), B (residues 29 for synapsin 1 or residues 30 for synapsin 2 to residues 112 in rat), and C (residues 113 to 420 in rat). Their tails are more variable and predominantly disordered in structure, including substantial intrinsically disordered regions (IDR) (Figure 2A). The N-terminus of synapsin 1 (domains A-C) interacts with SV phospholipids, while the proline-rich C-terminal end associates with protein components of the SV [21,22]. 

Quantitatively, on average 8.3 synapsin molecules are estimated to bind to each SV [23]. Given that many nerve terminals contain hundreds of SVs or more, this yields thousands of synapsin molecules in a typical nerve terminal. Indeed, the bulk concentration of synapsins within a nerve terminal has been estimated to be on the order of 150 μM [24]. The average number of copies per terminal varies between synapsin isoforms, ranging from around 1000 copies of synapsin 2a to 9000 copies of synapsin 1b [24]. Synapsins are not uniformly distributed within a presynaptic terminal. Immunogold labeling shows that synapsins are more enriched in the distal cluster of SVs, which is believed to be the morphological correlate of the RP (see below), and become less abundant near the AZs, where the RRP is located [25,26,27,28,29]. In addition, synapsin isoform expression varies between different types of neurons [30,31].

Synapsins were initially discovered as a substrate for protein kinase A (PKA); subsequently, it has been found that synapsins are phosphorylated by numerous other protein kinases, including the Ca^2+^/calmodulin-dependent protein kinase II (CaMKII) (Figure 2A) [32]. Phosphorylation at certain sites (PKA/CaMKI and CaMKII sites) causes synapsins to dissociate from SVs [21,33,34,35,36,37,38]. During repetitive synaptic activity, phosphorylation of synapsin 1a regulates mobilization of SVs to the RRP, as well as the transient dispersion of this protein out of nerve terminals and into adjacent axonal compartments [39,40]. Mutations that disable phosphorylation on either the PKA/CaMKI site or CaMKII sites of synapsin 1a slow down SV mobilization from the RP [40]. Presumably other isoforms behave like synapsin 1a, because the PKA phosphorylation site in domain A is conserved across all major isoforms (Figure 2A). In contrast, other isoforms do not possess CaMKII phosphorylation sites and presumably are not directly modulated by CaMKII. Therefore, while synapsin 1a and 1b are able to release SVs from the RP in response to CaMKII phosphorylation, the other isoforms must rely on PKA (or CaMKI) phosphorylation to mobilize SVs.

**Figure 2 cells-10-00658-f002:**
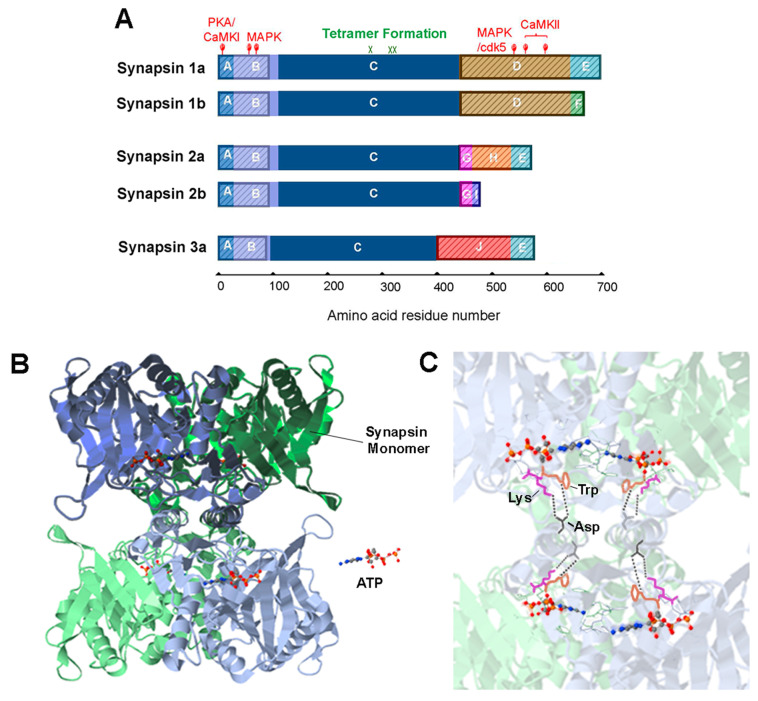
Synapsin domain structure and tetrameric structure of synapsins. (**A**). Synapsin isoforms consist of numerous domains, indicated by colors and letters. IDR regions are indicated by shading. Adapted from [41]. (**B**). Tetrameric structure formed by the C domain of synapsin 2 in the presence of ATP. Two synapsin monomers in a dimer—colored blue or green—are shaded with dark or light colors. A tetramer consists of two dimers. (**C**). The key residues mediating tetramer formation: aspartate (Asp, black), tryptophan (Trp, orange) and lysine (Lys, magenta); bonds between synapsin dimers are indicated by dashed lines. Structures in (**B**,**C**) adapted from data in [42], (accessible at PDB ID: 1i7l), visualized with the NGL viewer [43].

## Synapsins Maintain the SV Reserve Pool

A clear functional connection between synapsins and the RP was initially established by microinjection experiments that perturbed synapsin function. The first key result came from experiments done in lamprey giant axons, where a distal cluster of SVs become dispersed after injecting an antibody directed against domain E of synapsin 1a [6]. The injection also attenuated neurotransmitter release elicited by trains of action potentials that elicit sustained fusion of SVs, suggesting a diminished supply of SVs from the RP [6]. This study concluded that (1) the distal SV cluster serves as a RP; and (2) depletion of this cluster reduces the functional RP needed to sustain synaptic transmission during repetitive synaptic activity. Follow-up studies at the squid giant synapse tightened up the correlation between the distal SV cloud and the function of the RP. Short peptides from several synapsin domains were microinjected while measuring the rate of synaptic depression evoked by high-frequency activity, which monitors how quickly the SVs available for release in the RRP are depleted and thus provides a physiological monitor of SV mobilization from the RP to the RRP (Figure 3A,B) [44,45]. In brief, only peptides that hastened the rate of depression - indicating impairment of the RP – also depleted the distal SV cluster [44,45]. 

Genetic manipulations in mice have yielded results very similar to those of microinjection experiments [15,16]. Because of the existence of 3 synapsin genes, the clearest results have come from analysis of synapsin triple-knock-out (TKO) mice where all 3 genes are deleted [15]. Excitatory synapses and inhibitory synapses play opposite roles in neuronal circuits; they employ different neurotransmitters—often glutamate for central excitatory synapses and GABA for central inhibitory synapses—and are distinct in their molecular and cellular organization [46,47]. In excitatory synapses of cultured neurons from TKO mice, the number of SVs in the distal cluster again is reduced significantly while SVs docked at the AZ are intact (Figure 3C). While postsynaptic responses to single presynaptic action potentials are unaffected in TKO neurons, these neurons exhibit a 3-fold acceleration of synaptic depression during repetitive activity (Figure 3D). Taken together, the results indicate that complete loss of synapsins causes a selective impairment of the RP [15]. Further, rescue experiments indicate that only synapsin 2a is able to significantly restore the impaired RP [48]. In contrast, in inhibitory synapses from TKO mice, perturbation of synapsin molecules leads to a general reduction of SVs in both the RP and the RRP, but does not affect the rate of synaptic depression [15]. Instead, loss of synapsins seems to desynchronize GABA released in response to individual action potentials [31]. Moreover, all synapsins can rescue the physiological phenotype of TKO inhibitory synapses, except that synapsin 3a does not restore the desynchronization of GABA release [31]. In summary, it is clear that synapsins play different roles in excitatory and inhibitory synapses. However, the primary role of synapsins at excitatory synapses is to maintain SVs within the RP. Synapsins are also known to participate in other synaptic processes, including neuronal development, transmitter release probability, and short-term plasticity [31,49,50]. It is also important to note that even in the absence of synapsins, there are still some SVs within the RP, indicating likely roles for additional proteins.

## Models for Synapsin-Dependent Clustering of SVs in the RP

Although there is ample evidence that synapsins maintain glutamatergic SVs within the large, distal RP cluster, how they do this remains a puzzle. Below we describe and compare several models that have been proposed for the ability of synapsins to maintain SVs within the RP.

### Actin Scaffold Model

The original model, proposed by Paul Greengard and colleagues [19], is that binding of synapsins to both actin and to SVs serves to maintain SVs within the RP. In this model, synapsins tether SVs to the actin cytoskeleton, gathering the SVs on actin filaments and thereby forming the SV cluster. This model is consistent with experimental observations that synapsins can bind to both SVs and to actin filaments; indeed, synapsins can even promote actin polymerization [19]. Further, a variety of cytoskeleton filaments have been observed to associate with SVs [51]. However, several lines of evidence argue against this model. First, preventing actin polymerization in cultured hippocampal neurons does not dissipate the RP cluster [29,52]. Second, there is a paucity of actin-like filaments associated with the distal SV cluster [53]. Instead, electron microscopy (EM) analysis indicates that actin cytoskeleton filaments are prominently positioned at the peri-AZ region and are probably involved in recycling of SVs, providing a scaffold to guide recycled SVs back to the AZ [52,54]. In summary, it appears that actin is not a major contributor to RP maintenance. Instead, actin may indirectly affect the RP by concentrating synapsins within the presynaptic terminal [52], and thereby contribute to attracting SVs to the RRP [10]. Thus, the actin scaffold model will not be further considered in our review.

### Synaptic Vesicle Crosslinking 

A second model proposes that synapsins maintain the RP by crosslinking SVs (Figure 4A). This ***SV crosslinking*** hypothesis was inspired by discovery of short tethers that connect neighboring SVs; it proposes that these tethers interconnect SVs to form a network of SVs that constitute the RP cluster [53,55,56,57]. These tethers range from 30 to 60 nm in length, which is similar to the 47 nm length of isolated synapsin 1 molecules (which are composed of a 14 nm diameter head and a 33 nm tail) [51]. This similarity in size suggests that synapsins could serve as the SV tethers. Further, incubating SVs with synapsin 1 produces tethers between SVs in vitro [51], while knock-out of the synapsin 1 gene reduces the number of tethers between SVs in cerebellar mossy fiber terminals in vivo [58]. However, some tethers remain between SVs even in synapsin TKO neurons, albeit reduced in number [55]. Finally, increasing protein phosphorylation, by treatment of synapses with a phosphatase inhibitor, decreases tethers between SVs distal to the AZ [53]. Although such treatment could affect the phosphorylation state of many proteins, this result is consistent with previous observations that phosphorylation decreases binding of synapsins to SVs.

This model relies on synapsins binding not only to SVs, but also binding to each other to enable cross-linking of SVs. Indeed, it is well established that synapsins can form both homo- and heterodimers [42,59]. The primary site of inter-molecular binding is the highly conserved C domain of synapsins [42,59], although domain E may also play some role [60]. Peptides from domains C or E inhibit dimer formation [60], providing a possible mechanism for the ability of these peptides to disrupt the RP when microinjected into presynaptic terminals [44,45]. In the presence of ATP, domain C mediates formation of synapsin tetramers (Figure 2B); in rodent synapsin 1, this depends on coordination of three key residues, aspartate residue 290 (D290), tryptophan 335 (W335), and lysine 336 (K336) [42] (Figure 2C). These residues are highly conserved across all synapsin isoforms (Figure 2A), suggesting an ability to form either homo-oligomers or hetero-oligomers [42]. In vitro evidence indicates that adding ATP to SVs containing synapsin 1 and other synapsin isoforms promotes clustering of SVs and increases synapsin 1 tetramerization, suggesting that synapsin tetramers promote SV clustering [25]. However, to date there have been few clear-cut experimental tests of this model in living presynaptic terminals. It will also be important to distinguish the functions of synapsin homo-oligomers and hetero-oligomers, and whether these have similar or divergent roles in RP formation.

In summary, the SV crosslinking hypothesis relies on the presence of tether structures that appear to bridge SVs. However, the molecular composition of these tethers remains a question. While current evidence suggests that synapsins may serve as inter-vesicle tethers, this has not been established definitively [55]. While observations that synapsins can form oligomers in vitro provide a possible molecular substrate for SV cross-linking by tethers, the physiological role of synapsin oligomers is currently unknown. It has also been suggested that synapsins help to stabilize, rather than form, the tethers; this is supported by the observation that knocking out *SYN 1 & 2* reduces, but does not eliminate, formation of bridges between SVs [61]. While further investigation will be needed to definitively identify the role of synapsins in inter-SV tethers, at present no experimental evidence definitively excludes the SV crosslinking hypothesis.

### Liquid-Liquid Phase Separation 

Recently, it has been appreciated that interactions between multivalent biological macromolecules (e.g., proteins, RNAs) can result in the formation of liquid condensates that are distinct from the bulk cytosol, similar to the separation of oil droplets in a mixture of oil and water. Such liquid-liquid phase separation is now known to be involved in many cellular compartmentalization phenomena [62,63,64,65]. Formation of liquid condensates relies on IDR of proteins, which often consist of proline-rich sequences [65]. As the tail region of synapsins contain such an IDR (Figure 2A), Milovanovic et al. proposed that synapsin molecules can condense into distinct liquid phase droplets, and thereby trap SVs into the distal RP cluster [66] (Figure 4B). Experiments by these authors provided some encouraging initial support for this ***liquid-liquid phase separation*** model. They showed that isolated synapsin 1 molecules are able to phase separate in an aqueous solution in vitro [66]. Importantly, the resulting condensates are capable of clustering SV-like liposomes, mimicking the cluster of SVs within the RP [66]. Addition of CaMKII, a protein kinase that phosphorylates synapsin 1 (Figure 2A), dissolves the synapsin condensates and disperses clustered liposomes, reminiscent of the phosphorylation-dependent mobilization of the RP in neurons [66] (Figure 4B).

The liquid phase separation hypothesis is attractive because it can predict several presynaptic phenomena. For example, the random mixing of SVs expected within the synapsin condensate is consistent with evidence for such behavior in recycled SVs [25,67]. The exceedingly high concentration of synapsins measured within presynaptic terminals also is consistent with the high concentration of synapsin required for liquid-liquid phase separation [24,68]. 

Although biochemical assays clearly show that synapsins can form condensed droplets in vitro—and have led some to conclude that SV clustering can only arise from liquid-liquid phase separation by synapsins [68]—to date there have been few tests of this model in living neurons. An initial test has come from microinjection of an antibody generated against the synapsin IDR into the lamprey axon [69]. EM showed that this antibody dispersed the distal SV cluster, whether or not synapses were activated during antibody microinjection. This contrasts with a previous study showing that microinjecting a synapsin domain E antibody only affected the distal SV cluster within active synapses [6]. While dispersion of the clustered distal SVs by the IDR antibody is consistent with the model, it is at odds with the observation that this antibody promotes liquid phase formation in vitro [69]. Moreover, in excitatory TKO neurons, expression of synapsin 1a with an intact IDR domain is not able to rescue the impaired RP, which this model cannot explain [48].

Several important points related to the liquid-liquid phase separation model remain to be experimentally tested. First, there is still no direct observation of synapsin liquid condensates in living presynaptic terminals. Once detected, it would be valuable to determine whether deletion of the IDR domain produces any effect on the condensates or on SV clustering. Identifying synapsin condensates in live presynaptic terminals is important because other AZ components are independently capable of generating liquid-liquid phase separation; in principle, these could also contribute to SV clustering [70,71]. Given the very limited volume of a presynaptic terminal, it is unclear how synapsin droplets could interact with droplets formed by other proteins. Although Wu et al. showed that synapsin droplets can remain separate from the condensates of another AZ protein [72], the concentration of proteins used in their study was unphysiological and the size of the droplets formed was inconsistent with the dimensions of most central presynaptic terminals (around 1 μm in diameter). The roles of other presynaptic proteins must also be considered: for example, while expression of synapsin alone in COS7 cells does not yield liquid-liquid condensates, co-expression of synaptophysin does enable droplet formation [73]. Further, the time required to condense synapsin into droplets in vitro (around 50 min) is much longer than relevant physiological phenomena, such as the time course of synaptic vesicle recycling [66] or synapsin translocation during synaptic activity [39]. Mimicking the density of cellular cytoplasm by adding the crowding agent, polyethylene glycol, accelerates the speed of droplet formation [66]; however, it is hard to know how faithfully this condition simulates the true cytoplasmic environment. Therefore, whether condensed synapsin droplets are able to form rapidly and dynamically inside neurons – either alone or in cooperation with other presynaptic proteins - remains a crucial question. This is particularly important because protein composition determines the thermodynamics of phase separation [74]. Finally, while the effects of CaMKII suggest that phosphorylation of the variable C-terminal IDR of synapsin 1 can control phase separation and SV clustering, the conserved A domain on the N-terminal end of synapsins possesses a short disordered segment as well as a PKA/CaMKI phosphorylation site (Figure 2A). It is not clear whether this segment contributes to liquid phase separation, or whether PKA phosphorylation of this domain affects synapsin condensates (Figure 4B).

In conclusion, the *liquid-liquid phase separation* hypothesis is novel and is consistent with some of what we know about synapsins and the RP. However, more experimental work will be needed to establish the connection between the intriguing results obtained from reconstitution experiments and SV dynamics in living neurons.

## Limitations of the Two Models

The SV crosslinking and liquid-liquid phase separation models both intend to explain the mechanism responsible for clustering SVs within the RP and to explain the phosphorylation-dependent mobilization of SVs from the RP during synaptic activity. However, both models capture only part of the role of synapsins within neurons.

Firstly, the two hypotheses do not explain the heterogeneous functions of different synapsin isoforms. It has been clearly shown that synapsin 2a is the only major isoform that can rescue the glutamatergic RP deficit in cultured synapsin TKO neurons; the other isoforms—in particular synapsins 1a/b—fail to rescue [48]. These isoforms also differ in their Ca^2+^ sensitivity: although domain C is highly conserved, synapsins 1, 2 and 3 still display differences in the Ca^2+^ dependence of their ATP binding and dimerization [75,76]. The possible role of such Ca^2+^ regulation in RP maintenance is not addressed by any model. Further, although many models propose a central role for CaMKII in mobilization of SVs from the RP [19], synapsins 1a/b are the only isoforms possessing a CaMKII phosphorylation site [41,77]. More refinements to both models are necessary to better account for the unique properties of different synapsin isoforms.

Secondly, the two hypotheses concentrate on two distinct parts of synapsin molecules. The SV crosslinking hypothesis largely relies on dimerization and tetramerization mediated by the conserved C domain, while the liquid phase hypothesis depends upon on the variable IDR domains at the C-terminal end (domains D–J). Therefore, the two regions could act synergistically to achieve the biological functions of synapsins, with the variable C-terminal ends of the different synapsin isoforms potentially imparting additional regulatory complexity. Thus, it is possible that both models are relevant to the RP function of synapsins.

Thirdly, neither model takes into account the roles of other presynaptic proteins that have been implicated in RP maintenance. Such proteins include intersectin [78], synaptotagmin-7 [79] and undoubtedly others [80].

Fourthly, although synapsins have multiple roles in presynaptic terminals, both models primarily focus on clustering of SVs within the RP. Among the other synapsin functions to be explained are the following: (1) Synapsin molecules are preferentially targeted to presynaptic boutons; indeed synapsins are often used as molecular markers of presynaptic terminals [81,82]. There must be a mechanism that concentrates synapsins within the presynaptic bouton [82]. A structure-function analysis indicates the importance of domains B,C, and E in this process [82]. Because these regions overlap with the SV binding region of synapsins, targeting could simply arise from binding to SVs [82], though actin filaments are likely involved in this process as well [52]. Because synapsin 1b does not target well to presynaptic terminals [82], yet has an IDR similar to that of other isoforms, liquid phase separation cannot account for the inability of synapsin 1b to target properly. (2) Synapsins are also involved in the biogenesis or maintenance of SVs: the total number of SVs and levels of expression of SV-associated proteins are reduced by approximately half in TKO neurons [15]. It is unclear how loss of synapsins can lead to a reduction in SVs. (3) Neighboring synapses can share SVs with each other, which constitutes a SV superpool [83]. It is unclear how synapsins may participate in the sharing of SVs between the local RP and the superpool.

Lastly, the two models generalize the role of synapsins in RP formation without regard to whether the synapses are excitatory or inhibitory. Evidence suggesting that synapsins regulate SV pools differently in excitatory and inhibitory synapses, both in terms of neurotransmitter release and subcellular organization of SVs [15,31,48]. Specifically, TKO inhibitory neurons lose SVs both in their RRP and RP, while their excitatory counterparts have a selective loss of RP vesicles [15]. Further, these phenotypes are differentially rescued by different isoforms, as mentioned previously: while all synapsin isoforms rescue the phenotype of inhibitory neurons [31], only synapsin 2a rescues the phenotype of excitatory neurons [41,48]. These differences could arise from differential expression of synapsin isoforms, differences in the molecular organization of inhibitory and excitatory synapses, or even some synapsin interacting partners downstream of SV mobilization from RP. Whatever the cause of this important distinction between excitatory and inhibitory synapses, neither model is capable of explaining it.

## Perspectives

Given the evidence currently in hand, we cannot yet reach a conclusion regarding which model is correct. The SV crosslinking hypothesis focuses on morphological analysis of inter-vesicle tethers and biochemical measurements of synapsin oligomerization, while the liquid-liquid phase separation model is inspired by the biochemical properties of synapsins and the ability of synapsins to form condensed droplets under in vitro conditions. To distinguish between these models, future research will need to address the following questions:For the SV crosslinking hypothesis, which proteins constitute the inter-SV links? If they include synapsins, why do some links persist in TKO neurons? What is the relationship of the tethers to synapsin oligomers? Given the differences in RP formation by different synapsin isoforms in glutamatergic synapses, what differentiates the ability of these isoforms to form the RP?For the liquid phase separation hypothesis, how quickly can synapsin droplets form in neurons? How do regulatory protein kinases and phosphatases move in and out of the proposed synapsin condensate? How do the other synaptic proteins contribute to the liquid-liquid phase separation of synapsin?

The final question is whether the two models can be reconciled? Perhaps each model captures a different facet of the same process of SV clustering. Clearly, future studies will be necessary to address all of these questions.

## Figures and Tables

**Figure 1 cells-10-00658-f001:**
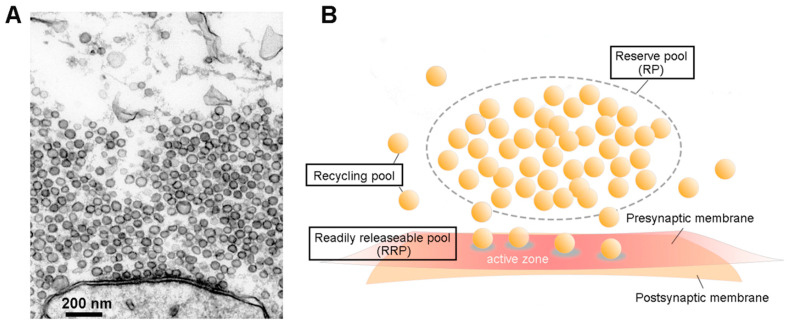
Synaptic vesicles (SVs) and their pools. (**A**). Electron micrograph of an active zone (AZ) from a squid giant synapse. SVs are the dark, membrane-bound circles that are both attached to the presynaptic plasma membrane and clustered nearby. (**B**). Classification of SV pools. A reserve pool (RP) occupies the distal volume of the presynaptic terminal, while a readily releasable pool (RRP) consists of vesicles docked at the AZ. The other freely moving SVs are assigned to a recycling pool.

**Figure 3 cells-10-00658-f003:**
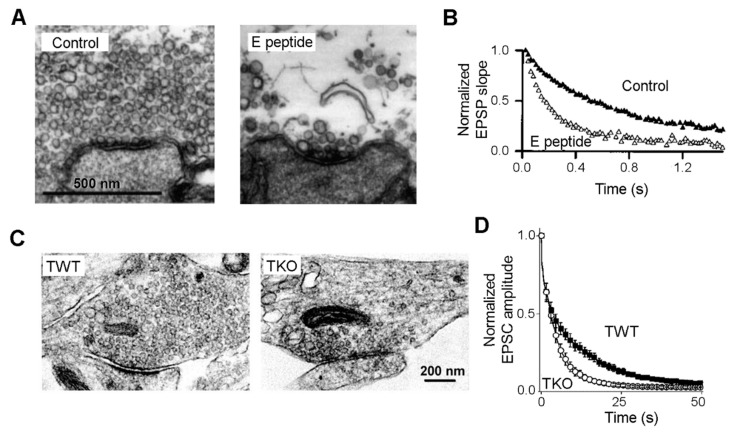
Synapsins and the RP: correlated structural and functional evidence. (**A**). Injection of a peptide from the E domain of synapsin into squid giant synapses disperses the distal cluster of SVs, while docked SVs remain intact. (**B**). Synaptic depression, evoked by a 50 Hz train of presynaptic action potentials, is hastened following peptide injection. (**A**,**B** adapted from [44]). (**C**). Comparison of excitatory synapses from synapsin triple-knockout (TKO) and wild-type (TWT) neurons. A distal cluster of SVs is absent in the TKO synapse. (**D**). Synaptic depression evoked by 10 Hz stimulation is accelerated in TKO neurons, in comparison to TWT neurons. (**C**,**D** adapted from [15]).

**Figure 4 cells-10-00658-f004:**
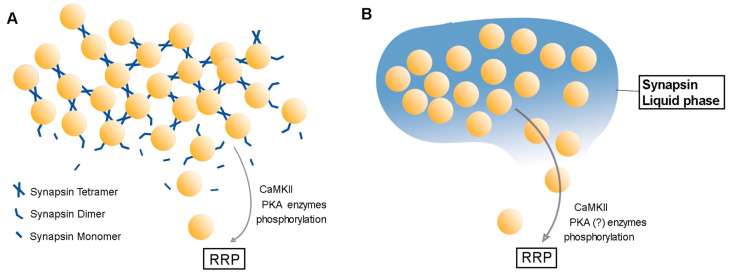
Reserve pool clustering models. (**A**). *Synaptic vesicle crosslinking* model. The vesicles are interlinked by tethers formed by synapsin oligomers. (**B**). *Liquid-liquid phase separation* model. Synapsin molecules together with SVs form a distinct liquid phase, trapping the SVs into a cluster.

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
