# Peer review of "Synapsins and the Synaptic Vesicle Reserve Pool: Floats or Anchors?"

_cells, 2021, doi:10.3390/cells10030658_

Round 1
Reviewer 1 Report
The role of synapsins, a very abundant protein in neurons particularly at presynaptic terminals has not been very clear and is the subject of this review/perspective manuscript. In the article the authors focus on the role of synapsins at maintaining/forming/mobilizing the reserve pool of synaptic vesicles. The article is very well written and seeks to not only summarize the findings to date but to propose a possible assimilation of two currently popular models/hypotheses about the role of synapsins in transmitter release.
A general comment is that there is no clear definition of what the reserve pool (and the RRP) actually is. Indeed the very existence of a separate reserve pool has been questioned and may be an artifact of the immature state of presynaptic boutons, at least in cultured mammalian neurons (Rose et al., Neuron. 2013 77(6):1109-21. doi: 10.1016/j.neuron.2013.01.021) and if there really are distinct pools they are not physically well-separated and the relative pool sizes can be extremely variable (e.g. Rey et al., 2020 30:2006-2017.e3. doi: 10.1016/j.celrep.2020.01.051). That being said, no one will argue that how synaptic vesicles are clustered close to where they are needed, and how their movement to the presynaptic membrane is directed (and whether it is directed) are highly important processes to understand.
This review is an excellent synthesis of the evidence for and against a key role of synapsins and the perspectives will I'm sure be appreciated by many readers. One more general comment. By the end one has the impression that the authors are suggesting that there are no alternatives to consider other than that synapsins are the controlers of vesicle clustering and mobiliization. Possibly this is not their intention and they might consider adding a sentence to the first paragraph of the 'perspectives' to remind the reader that there may be other important players.
minor:
line 13 evaluate is misspelled
Figure 1 legend Synaptic vesicles (SVs) ....'active zone (AZ)'
~line 65 text describing Figure 2 maybe should include CaMKI (PKA/CaMKI site as referred to line 227)
Figure 2 is the box labeled 'IDR' indicating a separate segment that is part of Synapsin 2b? In the figure I don't see the 'shading' referred to - ah maybe it is meant that the stripes/hatching indicate IDRs (just figured it out - the box should be the legend).
line 76 phosphorylation by(?) CaMKII
Figure 3. The E peptide seems to leave only docked and neither the RRP or RP remain - so are these really 'separate' pools?
Reviewer 2 Report
This manuscript is a succinct review on synapsins and its association with the synaptic vesicles (SV), especially the maintenance of the reserve pool of clustered SVs distal to the active zone (AZ). This reviewer’s concerns are mainly on the schematic diagrams which may mislead naïve readers.
Specifically,
- Fig.1, the diagram in B placed several SVs representing the “recycling pool” between the active zone and the “reserve pool”. Is this the intended message of the authors? If so, references need to be carefully cited. In this reviewer’s mind, the recycling pool could scatter around the cluster of SV (in view of the distribution of clathrin-coated vesicles).
Also, the diagram in B placed the clusters of SV far away from the active zone. This representation could be exaggerated and is not matching with the EM images shown in A.
The diagram in B suggests that the recycling pool of SV do not have synapsins associated with it. If so, this also needs to be cited with reference in the figure legend (like in Fig. 2).
- Fig. 4, citations should be included in the legend for easy reference.
Again, the same concern here as for the diagram in 1B, the clustered SV (the reserve pool) are placed too far away from the active zone. These drawings are NOT matching the EM images depicted in Fig. 1A or in Fig. 3, where there is no separation between the readily release pool of docked SV and the reserve pool.
Also, the placement of CaMKII molecules may be misleading. I suggest that the authors refer to (Tao-Cheng et al., 2006, Brain Cell biology 35: 117-124) that CaMKII is mostly lacking near the active zone under resting conditions, and that CaMKII molecules typically surround the SV clusters.
- The fact that EM evidence shows synapsins avoiding the active zone [ref #26], and that distribution of active zone cytomatrix proteins, bassoon and piccolo, are mutually exclusive to synapsins under resting conditions could be cited to support the phase separation model.
Reviewer 3 Report
Authors described synapsins and its function as a key protein for maintaining SVs within the reserve pool and discussed the potential models and mechanisms by which synapsins cluster SVs. The manuscript is well written and focused on the topic. In addition, the manuscript raised several very interesting questions:
Main points:
- Neurotransmission should be a rapid, precisely controlled, highly dynamic process. Authors gave a quantitative percentage of SVs in three different pools. Does the percentage of SVs in these three pools remain similar among different type of synapses or in different state of activities?
- Once SVs in the reserving pool being mobilized, where do new SVs come from and join the existing SV clusters? Being specifically transported through denovo synthesis or recycled SVs? SVs randomly moved and clustered among themselves or captured by existing SV clusters?
- Authors discussed actin-based cytoskeleton model of RV dynamics. How about other key cytoskeleton, such as microtubule, that usually plays a role in the transportation of membrane vesicles including synaptic vesicles and might also play a role in RV clustering?
- Authors mentioned a polarized localization of synapsins in SVs, more in distal side of SVs. Does the polarized localization of synapsins bring maintained throughout the process of neurotransmission? Or during mobilization? How about in the recycling SVs?
- Recycling pool occupies 10-15% of total SVs and supports rapid and repeated rounds of release of SVs. Does the SVs in the recycling pool and/or the released and recycled SVs goes back to SV clusters in the reserving pool?
- Do other key proteins and membrane components such as phospholipids in synaptic vesicles contribute to SV clustering?
Minor points:
- Certain terms might require explanation for the readers who are not very familiar with neurotransmission, such as trains of action potentials, the rate of depression et al.
- Synaptic depression and high-frequency activity provoked synaptic depression as a physiological monitor of SV mobilization need explanation.
- A general explanation of excitatory and inhibitory synapses and relevant examples will be helpful to understand the phenotypic changes in the synapsin TKO cells mentioned later.
